# Escape and Over-Activation of Innate Immune Responses by SARS-CoV-2: Two Faces of a Coin

**DOI:** 10.3390/v14030530

**Published:** 2022-03-04

**Authors:** Sameer-ul-Salam Mattoo, Seong-Jun Kim, Dae-Gyun Ahn, Jinjong Myoung

**Affiliations:** 1Department of Bioactive Material Science, Korea Zoonosis Research Institute and Genetic Engineering Research Institute, Jeonbuk National University, Jeonju 54531, Korea; drsameerulsalam@gmail.com; 2Center for Convergent Research of Emerging Virus Infection, Korea Research Institute of Chemical Technology, Daejeon 34114, Korea; sekim@krict.re.kr (S.-J.K.); dgahn@krict.re.kr (D.-G.A.)

**Keywords:** SARS-CoV-2, innate immunity, interferon, inflammasome

## Abstract

In the past 20 years, coronaviruses (CoVs), including SARS-CoV-1, MERS-CoV, and SARS-CoV-2, have rapidly evolved and emerged in the human population. The innate immune system is the first line of defense against invading pathogens. Multiple host cellular receptors can trigger the innate immune system to eliminate invading pathogens. However, these CoVs have acquired strategies to evade innate immune responses by avoiding recognition by host sensors, leading to impaired interferon (IFN) production and antagonizing of the IFN signaling pathways. In contrast, the dysregulated induction of inflammasomes, leading to uncontrolled production of IL-1 family cytokines (IL-1β and IL-18) and pyroptosis, has been associated with COVID-19 pathogenesis. This review summarizes innate immune evasion strategies employed by SARS-CoV-1 and MERS-CoV in brief and SARS-CoV-2 in more detail. In addition, we outline potential mechanisms of inflammasome activation and evasion and their impact on disease prognosis.

## 1. Introduction

*Coronaviridae*, a family of enveloped single-stranded RNA viruses, consists of two sub-families, *Orthocoronavirinae* and *Letovirinae* [1,2]. SARS-CoV-1, MERS-CoV, and SARS-CoV-2 belong to the sub-family *Orthocoronavirinae* and genus *Betacoronavirus*. Specifically, SARS-CoV-1 and SARS-CoV-2 belong to the subgenus *Sarbecovirus,* while MERS-CoV is classified under the subgenus *Merbecovirus* [3]. SARS-CoV-2 shares around 50% and 80% genomic homology with MERS-CoV and SARS-CoV-1, respectively [4]. Typically, their genome consists of non-structural, structural, and accessory proteins flanked by a 5′-cap and a 3′-poly (A) tail. Open reading frame (ORF)1a/b occupies two-thirds of the viral genome at the 5′ end and is translated into two large polyproteins, pp1a and pp1ab, which are then processed into 16 non-structural proteins (NSP1–NSP16) by two viral proteases, NSP3 (papain-like protease) and NSP5 (3C-like protease). The remaining one-third of the viral genome at the 3′ end encodes viral structural proteins (spike, S; envelope, E; membrane, M; and nucleocapsid, N) and accessory proteins. Various CoVs possess different numbers of accessory proteins. For example, eight accessory proteins (ORF3a, ORF3b, ORF6, ORF7a, ORF7b, ORF8a, ORF8b, and ORF9b) of SARS-CoV-1, five accessory proteins (ORF3a, ORF4a, ORF4b, ORF5, and ORF8b) of MERS-CoV, and nine accessory proteins (ORF3a, ORF3b, ORF6, ORF7a, ORF7b, ORF8, ORF9b, ORF9c, and ORF10) of SARS-CoV-2 have been established so far (Figure 1).

A timely and well-coordinated immune response against invading pathogens is necessary for the individual’s cell homeostasis, recovery, or survival. Host responses are triggered by pattern recognition receptors (PRRs) and germline-encoded cellular receptors, which recognize pathogen-associated molecular patterns (PAMPs) and danger-associated molecular patterns (DAMPs). A number of PRRs have been identified, among them, Toll-like receptors (TLRs) and retinoic acid-inducible gene I (RIG-I)-like receptors (RLRs) have been widely studied. Most of the viral proteins, including structural, NSP, and accessory proteins, have been reported to play differential roles in innate immune evasion. In contrast, the activation of inflammasomes by CoV initiates an inflammatory form of cell death, triggering the release of proinflammatory cytokines IL-1β and IL-18 [5,6,7,8,9]. Uncontrolled inflammasome activation has been associated with disease severity [10].

This review summarizes the innate immune evasion strategies employed by SARS-CoV-1 and MERS-CoV with extensive focus on SARS-CoV-2. Moreover, we highlight the activation mechanisms of inflammasome and its role in the pathophysiology of the COVID-19.

## 2. Host Innate Immune Sensors

RLRs are cytosolic sensors of viral RNA that respond by inducing the production of type I IFNs and other pro-inflammatory cytokines [11] (Figure 2).

RLRs can recognize dsRNA and RNA-DNA hetero-duplexes, which include the genomes of dsRNA viruses and RNA transcripts of RNA and DNA viruses. The two well-characterized RLRs are RIG-I and melanoma differentiation-associated gene 5 (MDA5). RIG-I recognizes RNA with a 5′ triphosphate moiety (absent in cytosolic RNAs of mammalian host cells), and MDA5 recognizes long dsRNA (1–6 kb), which is longer than average mammalian RNAs [12,13]. Upon viral recognition, RIG-I and MDA5 interact with the mitochondrial antiviral signaling protein (MAVS), resulting in the formation of prion-like aggregates, which is essential for the biological functioning of MAVS [14]. The activation of MAVS results in the recruitment of TNF receptor-associated factor (TRAF) 3/6 and inhibitor of NF-κB kinase (IKK) family members (IKKε, TBK1, IKKα/β) for the activation/phosphorylation of IFN regulatory factors 3 and 7 (IRF3/7) as well as nuclear transcription factor-κB (NF-κB) [15]. IRF3/7 and NF-κB eventually translocate to the nucleus, inducing the expression of pro-inflammatory cytokines, such as IFNs. Once secreted, IFNs act in an autocrine and paracrine manner to induce the expression of IFN-stimulated genes (ISGs) via JAK and STAT (JAK-STAT) signaling pathways [16]. The binding of type I IFNs to its receptor, which is composed of at least two different subunits, including IFN-alpha/beta receptor (IFNAR1) and IFNAR2, results in a rapid auto-phosphorylation and activation of receptor-associated TYK2 and JAK1 [17]. Eventually, a heterotrimeric transcription factor called IFN-stimulated gene factor 3 (ISGF3) is activated, consisting of an SH2-phosphotyrosine-mediated heterodimer of STAT1 and STAT2 in association with IFN regulatory factor 9 (IRF9). The ISGF3 complex undergoes nuclear translocation and binds IFN-stimulated response elements (ISREs) present in the promoter of certain ISGs, eventually initiating the expression of ISGs [17,18,19,20]. In the context of SARS-CoV-2, it has been proposed that MDA5 and RIG-I recognize SARS-CoV-2 in cell type-specific manners [21,22,23,24]. For example, MDA5 was shown to recognize SARS-CoV-2 in human lung epithelial cells [24], while the RIG-I knockout HEK293T cell line did not show an IFN-β response upon SARS-CoV-2 RNA stimulation [25]. In addition, the Papain-like protease domain of NSP3 interacted with MDA5 and antagonized ISG15-dependent MDA5 activation through active de-ISGylation [26].

In humans, 10 TLRs have been recognized so far, which are expressed in the cell membrane or endosomes. Several of them are well-known to recognize RNAs of the invading pathogens: TLR3 senses dsRNA, while TLR7 and TLR8 recognize ssRNA (Figure 2). Upon stimulation, TLRs recruit cytosolic adapter molecules MYD88 and/or TRIF. These adapter molecules coordinate downstream signaling pathways by recruiting several ubiquitin ligases, such as TNF receptor-associated factor (TRAF) 3 and TRAF6 [27]. Ubiquitin ligases engage with antiviral kinases such as TANK-binding kinase 1 (TBK1), I-kappa-B kinase (IKKε), and the IKKα/β/γ complex. Subsequently, the transcription factors IRF3, IRF7, and NF-κB are activated, resulting in the production of type I IFNs and proinflammatory cytokines [27]. It was shown that TLR2 senses SARS-CoV-2 E protein to produce inflammatory cytokines [28], and TLR4 directly interacts with the S protein [29]. In addition, mutations in TLR3 and TLR7 might be playing a role in SARS-Co-V-2 disease outcomes: in-born errors in TLR3 and IRF7 were associated with disease severity and mortality in COVID-19 patients [30]. Another study identified two rare mutations in TLR7 in four young males with severe COVID-19 from two unrelated families [31]. Out of four, one patient died. These mutations were associated with transcriptionally downregulated downstream type I IFN signaling, as measured by significantly decreased mRNA expression of IRF7, Interferon Beta 1, and ISG15 on stimulation with imiquimod, a TLR7 agonist. Asano, et al. [32] reported that 1% of individuals with COVID-19 younger than 60 years who died had a TLR7 impairment, an uncommon disease that increases vulnerability to viral infections [32].

Inflammasomes are a group of multimeric protein complexes consisting of a sensor molecule, an adaptor protein ASC, and caspase 1 [33]. Inflammasome assembly is triggered by a variety of substances that emerge during infections, tissue damage, or metabolic imbalances. Once stimulated, activated caspase 1 proteolytically processes the pro-inflammatory cytokines interleukin-1β (IL-1β) and IL-18 [33]. Moreover, inflammatory caspases cleave the pyroptotic executor protein gasdermin D (GSDMD) into amino-terminal and carboxy-terminal fragments. The GSDMD amino-terminal fragment binds to acidic lipids, oligomerizes, and inserts itself into cell and organelle membranes to form sizable pores, resulting in the release of IL-1β and ΙL-18, which are strong inducers of downstream immune responses and various alarmins, eventually leading to pyroptosis [34,35,36]. In general, the NLRP3 inflammasome is fully activated (canonical activation) in two steps: a priming signal (signal 1) and an activation signal (signal 2) [37]. Priming has at least two functions: (a) The inflammasome components NLRP3, caspase 1, and pro-IL-1 are upregulated, triggered by the recognition of PAMPs or DAMPs by PRRs or through cytokines such as TNF-α and IL-1β (b) NLRP3 post-translational modifications (PTMs), including ubiquitylation, phosphorylation, and sumoylation, which stabilize NLRP3 in an auto-suppressed inactive but signal-competent state [37]. NLRP3 inflammasome assembly and activation is induced by multiple DAMPS and PAMPs [5,38]. PRRs, such as DExD/H-box RNA helicase family members DHX33 and DDX19A, have been reported to interact with and activate NLRP3 after sensing reoviral genomic RNA [38]. Moreover, the roles of ZBP1, RIG-I, and TLR3 have also been shown to be involved in driving the activation of the NLRP3 inflammasome [38]. In the context of SARS-CoV-2, an increasing body of evidence supports the role of dysregulated inflammasome activation and proinflammatory cytokines in acute lung injury upon SARS-CoV-2 infection (Figure 3) [39].

As already discussed, CoVs suppress an early type I IFN synthesis and signaling to allow unrestrained viral replication. The lysis of infected cells in the lung can release alarmins to activate inflammasome. NLRP3 is expressed in myeloid and lymphoid-origin immune cells and alveolar epithelial and pulmonary endothelial cells, where its over-activation can lead to lung injury [11]. NLRP3 inflammasome activation by CoVs has been proposed to occur by other mechanisms as well: K^+^ efflux or Ca^2+^ influx induced by Viroporins including E, ORF3a, and ORF8 [40,41,42,43,44]. Viroporins are small hydrophobic proteins encoded by viruses that oligomerize in the membrane of host cells, forming hydrophilic pores [45].

## 3. Innate Immune Escape by CoVs

### 3.1. Sheltering and Modifying Viral RNA to Evade PRR Recognition

NSP3, NSP4, and NSP6 of CoVs have been reported to play a critical role in hijacking the ER membrane to form double-membrane vesicles, where the virus replicates to evade cytosolic PRR sensing [46,47,48]. Further, viral RNAs undergo post-transcriptional modification to resemble eukaryotic RNA to escape the recognition by host sensors. For example, RNA capping, which involves several enzymatic steps, is carried out by viral NSPs. NSP13 hydrolyses the first phosphate from the nascent RNA, followed by the transfer of a guanosine monophosphate by an unknown enzyme [49]. Then, the NSP10/14 heterodimer transfers a methyl group from S-adenosylmethionine to the guanidine N7 to form Cap-0-RNA [50]. Eventually, the NSP16/NSP10 heterodimer methylates the ribose 2′-O of the first nucleotide of the nascent viral Cap-0-RNA [51,52]. The 2′-O methylation of the Cap-0-RNA prevents activation of cytoplasmic sensors, MDA5, and RIG-I [52,53,54].

### 3.2. Inhibiting Host Protein Synthesis and the Degradation of Proteins

NSP1 has been shown to play an inhibitory role in protein synthesis. SARS-CoV-2 NSP1 associates with the host mRNA export receptor NXF1-NXT1, preventing NXF1 from adequately interacting with mRNA export adaptors [55]. Eventually, numerous cellular mRNAs are trapped in the nucleus. In addition, SARS-CoV-1 and SARS-CoV-2 NSP1 interact with the 40S ribosomal subunit and strongly block host protein translation. It has been revealed that N- and C-terminal regions of SARS-CoV-2 NSP1 block mRNA transport and the mRNA entry tunnel of ribosomes, respectively [55]. These findings suggest that NSP1 has evolved to target different phases in the mRNA biogenesis pathway. In addition, SARS-CoV-2 NSP14 was shown to target IFNAR1 for lysosomal degradation [46]. Further, SARS-CoV-2 NSP1 was shown to cause depletion of antiviral factors Tyk2 and STAT2, which may be due to the NSP1-mediated global reduction of translation [56]. Moreover, NSP14 of SARS-CoV-1, MERS-CoV, and SARS-CoV-2 has been reported to inhibit host translation [57,58]. Interestingly, SARS-CoV-2 NSP10 enhances the translation inhibition activity of NSP14, primarily by structural stabilization [58]. Through protein–protein interactions, NSP10 activates the enzymatic activity of NSP14 and NSP16 [58,59,60]. It was further shown that SARS-CoV-2 NSP16 binds mRNA recognition domains of U1 and U2 splicing RNAs and disrupts host mRNA splicing, resulting in decreased ISG activity [61]. In addition, the SARS-CoV-2 M protein was shown to degrade TBK1, via K48-linked ubiquitination [62]. Further, NSP3 of SARS-CoV-2 was shown to cleave IRF3 via its papain-like protease activity [63], while SARS-CoV-1 ORF9b targeted the MAVS signalosome leading to MAVS, TRAF3, and TRAF6 degradation [64].

Human inborn errors of immunity (IEI) is a catch-all name for primary immunodeficiency diseases (PIDs), which encompass a wide range of clinical symptoms, from infection susceptibility to immunological dysregulation and cancer. In a cohort of 94 patients with IEI who had SARS-CoV-2 infection, more than 30% had mild symptoms, and 10 patients were asymptomatic [65]. Eighteen patients required intensive care, and 9 of them died because of COVID-19. However, in a cohort of 121 COVID-19 patients with IEI, the disease outcome was mild in most patients, but the case fatality ratio (the proportion of people diagnosed with a certain disease who end up dying of it) was higher than in the general population [66]. Certain autoantibodies have been shown to increase susceptibility to infections mimicking IEI. For example, autoantibodies against type I IFNs have been supposed to play a role in COVID-19 disease prognosis. Although rare, high titers of neutralizing autoantibodies against type I IFN-α2 and IFN-ω were reported in about 10% of patients with severe COVID-19 [67], while those autoantibodies were neither detected in healthy people nor in infected patients with milder symptoms.

### 3.3. Impairing the Type I IFN Synthesis and Signaling Pathway

#### 3.3.1. NSPs

##### NSP1

NSP1 of CoVs has been reported to hamper innate immune responses by targeting multiple biological pathways. SARS-CoV-1 NSP1 has been reported to inhibit virus-induced dimerization of IRF3 and activation of the IFN-β promoter. SARS-CoV-2 NSP1 was shown to inhibit virus induced-expression of type I and type III IFNs as well as IFN-stimulated response elements (ISREs). Moreover, compared with SARS-CoV-1 and MERS-CoV, SARS-CoV-2 NSP1 suppressed STAT1 and STAT2 phosphorylation more efficiently [68].

##### NSP3 and NSP6

SARS-CoV-1 and SARS-CoV-2 NSP3 can bind IRF3 and prevent its phosphorylation, dimerization, or translocation to the nucleus, effectively inhibiting type I IFN synthesis [69]. SARS-CoV-1 NSP3 can also inhibit the NF-κB signaling pathway by stabilizing IκBα, an NF-κB inhibitor. It has been reported that NSP3 antagonizes ISGylation and ubiquitination of host antiviral proteins, resulting in suppressed antiviral responses.

NSP6 disrupts the host’s immunological response by interacting with TBK1 to inhibit IRF3 phosphorylation without impairing TBK1 phosphorylation [68]. Moreover, NSP6 inhibited phosphorylation of both STAT1 and STAT2 [68]. Of note, SARS-CoV-2 NSP6 inhibited type I IFN production more efficiently than SARS-CoV-1 and MERS-CoV [68].

##### NSP7, NSP8, and NSP12

NSP12, also known as RNA-dependent RNA polymerase, is a key component of the virus replication and transcription complex [70]. NSP12 possesses high polymerase activity with the assistance of additional cofactors, NSP7 and NSP8 [70]. NSP8 inhibits type I IFN production by binding to MDA5 and impairing its K63-linked polyubiquitination [71]. In addition, both NSP8 and NSP9 were shown to bind 7SLRNA, a component of the signal recognition particle, and disrupt protein trafficking [61]. NSP12 inhibited IFN-β promoter activation triggered by RIG-IN (an active form of RIG-I), MDA5, MAVS, and IRF3-5D (a constitutively active IRF3 mutant) [72]. Moreover, NSP12 suppressed the nuclear translocation of IRF3 [72].

##### NSP13

It has been reported that NSP13 suppresses type I IFN production [68,73]. Mechanistically, NSP13 was shown to interact with TBK1 and hamper the association of TBK1 with TRAFs and the consequent recruitment of TBK1 to MAVS, eventually suppressing type I IFN production [73]. In addition, NSP13 also suppressed type I IFN signaling by impeding STAT1 and STAT2 phosphorylation, resulting in the retention of STAT1 in the cytoplasm and impeding stimulation of the ISRE promoter [68,74]. Moreover, NSP13 was shown to interact with the deubiquitinase USP13, which deubiquitinates and stabilizes NSP13 [73].

##### NSP14

As already stated, SARS-CoV-2 NSP14 plays a vital role in evading PRR recognition and further causes host protein degradation and interferes with the host translation system. In addition, it was shown that NSP14 inhibits nuclear localization of IRF3 [75]. SARS-CoV-2 NSP14 inhibited RIG-I-mediated IFN-β promoter activity [75].

#### 3.3.2. Structural Proteins

##### N Protein

The CoV N protein binds the SPRY domain of TRIM25, preventing TRIM25-mediated RIG-I ubiquitination and activation, thus inhibiting the production of type I IFNs [76,77,78]. In addition, SARS-CoV-2 has been shown to inhibit TBK1-IRF3 interaction and phosphorylation and nuclear translocation of IRF3 induced by poly (I:C) or RNA viruses. The dimerization domain of the N protein, required for its liquid phase separation with viral RNA, inhibited Lys63-linked polyubiquitination and aggregation of MAVS, thus repressing the innate antiviral immune responses [79]. In addition, the SARS-CoV-2 N protein was reported to suppress the phosphorylation and nuclear translocation of STAT1 and STAT2 [80]. According to a recent article, it is worth noting that the SARS-CoV-2 N protein may have a dual function [81]. A low concentration of the N protein acted as an IFN antagonist, while a high dose strongly induced type I IFN and inflammation.

##### M Protein

The MERS-CoV M protein suppresses TBK1-dependent phosphorylation of IRF3. Tbe SARS-CoV-1 M protein physically associates with RIG-I, TBK1, TRAF3, and IKKε. The M protein sequesters some of these proteins in membrane-associated compartments, preventing TRAF3·TANK·TBK1/IKKϵ complex formation and thereby inhibiting TBK1/IKKϵ-dependent activation of IRF3/IRF7 transcription factors [82]. The SARS-CoV-2 M protein inhibited RIG-I—MAVS, MAVS—TBK1, TRAF3—TBK1 interactions and downregulated the phosphorylation and nuclear translocation of IRF3 [68,83]. Of note, the TM1-deficient M protein interacted with MAVS and TBK1 but not with RIG-I. Moreover, the M protein also inhibited MAVS aggregation [84].

#### 3.3.3. Accessory Proteins

##### ORF3a 

SARS-CoV-2 ORF3 inhibits IRF3 nuclear translocation and STAT1 phosphorylation [68,74]. SARS-CoV-1 ORF3a localizes to the plasma membrane of the endoplasmic reticulum (ER)-Golgi compartment. It induces ER stress by activating the PKR-like ER (PERK) kinase pathway, leading to the phosphorylation, ubiquitination, and lysosomal degradation of IFNAR1 [85].

##### ORF3b 

Compared to SARS-CoV-1, it was observed that SARS-CoV-2 ORF3b has a stronger inhibitory effect on the nuclear translocation of IRF3 and subsequent IFN-β production, presumably because of sequestering IRF outside the nucleus more efficiently [86]. Mechanistically, SARS-CoV-1 ORF3b has a nuclear localization signal (NLS) at the C-terminal and is equally distributed between the cytosol and nucleus, whereas SARS-CoV-2 ORF3b lacks the NLS and mainly resides in the cytosol [86,87]. Moreover, mutations in SARS-CoV-2 ORF3b have been presumed to be linked with pathogenicity. For example, a study screened 17,000 SARS-CoV-2 sequences and isolated a longer variant (Ecuador variant) of SARS-CoV-2 ORF3b in two critically ill patients [86]. The authors observed that the Ecuador variant ORF3b has enhanced anti-type I IFN activity more than parental SARS-CoV-2.

##### ORF6

It has been demonstrated that ORF6 encoded by both SARS-CoV-1 and -2 inhibits IRF3 and STAT nuclear translocation [68,88,89]. Mechanistically, ORF6 binds to importin karyopherin alpha 2 (KPNA2) to block IRF3 and STAT nuclear translocation, leading to the suppression of both type I IFN production and signaling [68,88]. In addition, it was shown that ORF6 localizes to the nuclear pore complex (NPC) and directly interacts with Nup98-Rae1 via its C-terminal domain to impair docking of the cargo-receptor (karyopherin/importin) complex and disrupts STAT nuclear translocation [89,90]. A methionine-to-arginine substitution at residue 58 impaired ORF6 binding to the Nup98-Rae1 complex and abolished its IFN antagonistic function [90]. In contrast to SARS-CoV, SARS-CoV-2 ORF6 may more dramatically suppress protein expression through a stronger interaction with the Nup98-Rae1 [89].

Another critical inhibitory role of SARS-CoV-2 ORF6 was shown in the downregulation of major histocompatibility complex class I (MHC-I) expression, whose expression also depends on type II IFNs (IFN-γ) [91]. MHC-I aids in antiviral immunity by facilitating antigen presentation to T cells. It has been demonstrated that T cells are also important for protection against SARS-CoV-2 [92]. Upon activation of the IFN-γ receptor by IFN-γ, STAT1 is phosphorylated, followed by dimerization and nuclear translocation. STAT1, as a transcription factor, initiates the expression of IFN-γ-inducible genes, including interferon responsive factor 1 (IRF1) and NOD-like receptor family CARD domain-containing five (NLRC5). After translation, IRF1 and NLRC5 translocate to the nucleus and function as transcription factors to induce the MHC-I pathway. It was shown that ORF6 downregulates MHC-I expression via its C-terminal domain by two pathways: (1) by impeding STAT1 nuclear import, resulting in diminished upregulation of NLRC5 and IRF1 gene expression, eventually limiting MHC-I expression; (2) by blocking karyopherin complex-dependent nuclear import of NLRC5, further attenuating MHC-I production [91].

##### ORF7

SARS-CoV-2 ORF7a contains a cytoplasmic di-lysine motif (KRKTE) for endoplasmic reticulum (ER) localization [93]. Both ORF7a and ORF7b strongly inhibited ISRE luciferase activity induced by recombinant IFN-α treatment [68]. ORF7a repressed STAT1 phosphorylation, whereas ORF7b inhibited the phosphorylation of both STAT1 and STAT2 transcription factors [68].

##### ORF9b

SARS-CoV-2 ORF9b was depicted to antagonize type I IFN pathway through its interaction with the translocase of the outer membrane, TOM70 [94]. Two mechanisms were proposed: (1) ORF9b may compete with Heat shock protein 90 (HSP90) for binding to TOM70. HSP90 physically interacts with TOM70 and plays a critical role in the response of TOM70-mediated IFN-I activation; (2) ORF9b may induce lactic acid production, which has been shown to inhibit type I IFN responses by interacting with TOM70. Moreover, SARS-CoV-2 ORF9b was demonstrated to disrupt K63-linked ubiquitination of IKKγ and inhibit IKKα/β/γ-NF-κB signaling [25].

### 3.4. Innate Immune Evasion by Other CoV Viral Components

MERS-CoV ORF8b, as well as ORF4a and ORF4b co-expression, significantly downregulated MDA5 protein levels [95]. In addition, ORF8b also inhibited TBK1-mediated induction of NF-κB [40].

In summary, the cellular targets and potential inhibitory mechanism(s) of SARS-CoV-2-encoded proteins are summarized in Table 1.

## 4. Inflammasome Activation by CoV 

The SARS-CoV-1 E protein embeds in the endoplasmic reticulum–Golgi intermediate compartment (ERGIC) membranes, causing a cytosolic Ca^+^ influx and activation of NLRP3, resulting in the overproduction of IL-1β in Vero epithelial cells, transiently transfected with inflammasome components: NLRP3, ASC, and procaspase-1 [41]. SARS-CoV-1 ORF3a-induced NLRP3 inflammasome activation required K^+^ efflux and mitochondrial reactive oxygen species [42] (signal 2). The SARS-CoV-1 ORF3a protein promoted p105 ubiquitination and processing, NF-κB activation, and pro–IL-1β gene transcription, leading to signal 1. Moreover, ORF3a associated with TRAF3 and ASC to induce K63-linked polyubiquitination of ASC, leading to signal 2 [43]. Of note, ORF3a alone was sufficient to stimulate both signal 1 and signal 2 for NLRP3 inflammasome activation [43]. SARS-CoV-1 ORF8a has been observed to form ion channels when reconstituted in artificial lipid bilayers at elevated temperature (38.5 °C) [44]. In addition, SARS-CoV-1 ORF8b has been shown to activate NLRP3 by providing a potent signal 2. Mechanistically ORF8b interacts directly with the LRR domain of NLRP3 and localizes with NLRP3 and ASC in cytosolic dot-like structures [45].

The overexpression of SARS-CoV-2 ORF3a caused K^+^ efflux, NLRP3 activation, and IL-1 β release in the lung epithelial cell line A549 [96]. Among structural proteins, the SARS-CoV-2 E protein has been observed to function as a K^+^-permeable cation channel, lysing cells and causing ARDS-like disease in mice [97]. In addition, he N protein of SARS-CoV-2 was able to bind NLRP3 directly, inducing inflammasome assembly and upregulating the expression of inflammatory factors, resulting in mouse lung injury and aggravating the death of mice in the sepsis model [98]. The processing and release of IL-18, which is dependent on the activation of the inflammasome and gasdermin D, has been linked with severe COVID-19 and has emerged as a highly predictive biomarker of mortality [6,7,8,9]. Inflammasome activation can be massively upregulated by a positive feedback loop, resulting in a cytokine storm and lung injury. For example, IL-1β binds to IL-1R and induces a NF-κB response, eventually resulting in enhanced transcription of pro-IL-1β, which can happen in myeloid cells recruited to the lung [99]. As a result of positive feedback, cytokine release and subsequent immune cell recruitment can be serially repeated, leading to the establishment of a hyperinflammatory milieu with a high load of cytokine-secreting monocytes and neutrophils, which is likely to have a role in severe lung damage. Moreover, IL-1β-mediated activation of endothelial cells and IL-6 secretion can downregulate vascular endothelial cadherin (VE-cadherin) transcription and increase the production of vascular endothelial growth factor (VEGF), respectively [100,101]. Downregulation of VE-cadherin results in the loss of adherens junctions, which are critical to barrier integrity, and the upregulation of VEGF weakens the pulmonary endothelium through VE-cadherin internalization. These events can compromise gas exchange by promoting the accumulation of interstitial and alveolar fluid [102,103]. In summary, positive feedback by inflammatory cytokines, followed by uncontrolled recruitment of immune cells, could lead to a cytokine storm and severe lung damage.

In contrast, Nucleocapsid binding to GSDMD was shown to suppress pyroptosis, preventing immunological signaling components from being secreted to the outside [104]. For example, GSDMD inhibition allowed host cells to accumulate large amounts of mature IL-1 in the cytoplasm, resulting in the release of a large amount of inflammatory cytokines in the lytic phase [104]. This strategy might help the virus to evade immune assault at early stages of infection and eventually result in a large quantities of cytokine release from dying monocytes at later stages, which would promote hyperinflammation and a cytokine storm in patients [104,105]. In addition, NSP1 and NSP13 were also shown to inhibit NLRP3-inflammasome-induced caspase-1 activity and IL-1β secretion [106]. In addition, NSP5 was observed to cleave nucleotide-binding leucine-rich repeats harboring receptors NLRP12 and TAB1, both of which are involved in the inflammatory response [63]. As most of these studies were conducted by the over-expression of viral proteins, pro- and anti-inflammasome activation by Coronavirus-encoded genes need to be evaluated under close scrutiny for their biological importance in the context of viral infections.

## 5. Therapeutics Targeting Innate Immune System

During the COVID-19 pandemic, various treatment strategies have been investigated, including antiviral therapies and inflammatory cytokine antagonists.

Antivirals, including remdesivir; molnupiravir; paxlovid; and anti-spike monoclonal antibodies, including casirivimab–imdevimab, bebtelovimab, and sotrovimab, have been approved by the FDA or have been given EUA [107]. In addition, antiviral therapies, including type I and type III IFN treatments, modulate immune cell activation, and inflammation has also been explored. For example, a preclinical study showed that exogenous type I IFN therapy prior to infection reduced the viral load [108]. However, the effect was limited once the infection was established. Similarly, a study showed that the administration of IFN-α2b during the early phase of COVID-19 (within 5 days post-hospitalization) was associated with a decreased mortality [109]. However, later administration of IFN-α2b was associated with an increase in mortality. In addition, a RIG-I agonist, stem-loop RNA 14, prevented viral infection in the lower respiratory tract and the development of severe disease in a mouse model of SARS-CoV-2 infection [110]. Similarly, like type I IFNs, it is well established that type III IFNs are the first-line defense against viral infections [111]. SARS-CoV-2 infection was reduced when human Calu-3 cells, intestinal epithelial cells, and colon organoids were pretreated with IFN- λ, while the absence of the type III IFN receptor increased susceptibility to infection [112,113,114]. Similarly, in mice, the administration of pegylated (PEG)-IFN-λ1a diminished SARS-CoV-2 replication [114]. Interestingly, in observational studies, increased IFN-λ1 and IFN-λ2 levels in serum were associated with a better prognosis [115,116]. Based on these data, clinical trials have begun to evaluate the effect of PEG-IFN-λ1a therapy in COVID19 patients, which have shown contradictory results. In a phase 2 trial, patients with the mild disease showed no significant differences in the time to symptom resolution or other clinical metrics between the PEG-IFN-λ1a and placebo groups [117]. However, another phase 2 trial study reported that PEG-IFN-λ1a decreased the levels of mid-turbinate viral load by day 7 [118].

In addition, inflammatory cytokine agonists, such as anti-IL-1 and anti-IL-6 therapies, have been evaluated. For example, with antagonists to IL-1 or its receptor, results are more or less contradictory: (1) an observational study found that in COVID-19 hospitalized patients, two doses of canakinumab, an anti-IL-1β antibody, can lead to clinical improvement, while a single dose was reported to be ineffective in a phase 3 study [119,120]; (2). anakinra, an IL-1 receptor antagonist, was shown ineffective in earlier studies [121]. Conversely, other studies observed a reduction in inflammatory markers and mortality in patients with COVID-19 [122,123]. To resolve apparent controversy in those studies, a more in-depth analysis of cytokine signaling inhibitors is warranted.

## 6. Conclusions

The innate immune system is the first line of defense against invading pathogens. Pathogens are recognized directly or indirectly by various PRRs, including TLRs, RLRs, inflammasome sensors, and so on. Downstream of sensing, IFN synthesis, signaling, cytokine production, and cell death are general aspects of the innate immune response that can inhibit viral replication by reducing viral gene expression and eliminating infected cells. CoVs have developed different strategies to escape recognition by innate immune sensors and downstream signaling pathways, which help efficient viral gene expression and genome amplification. The suppression of the Type I IFN synthesis or signaling pathway and inflammasome over-activation might play critical and diverse roles in COVID-19 disease prognosis. Although the protective role of inflammasome signaling and IL-1β release has been demonstrated against different pathogens, their uncontrolled activation might lead to enhanced tissue damage, along with unrestrained recruitment of immune cells to the site of infection. However, the exact mechanisms regulating the dominance and temporal association of these pathways is not fully understood. The administration of recombinant IFNs could restore an innate antiviral immune response, enabling the host immune system to control SARS-CoV-2 infection more efficiently and limiting subsequent hyperinflammation. However, the clinical trials have not reached a sound conclusion regarding the efficacy and proper timing of antiviral therapies. Non-structural, M, and N proteins, being potential holders of dominant T cell epitopes [124], shall be mapped precisely to elucidate the motifs responsible for innate immune evasion, which will pave the way for the development of more efficient and safe vaccines.

## Figures and Tables

**Figure 1 viruses-14-00530-f001:**
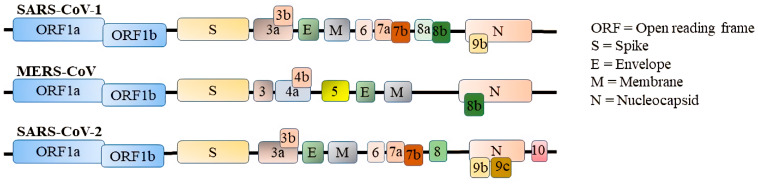
The CoVs structure, including ORF1a/b, structural proteins, and accessory proteins.

**Figure 2 viruses-14-00530-f002:**
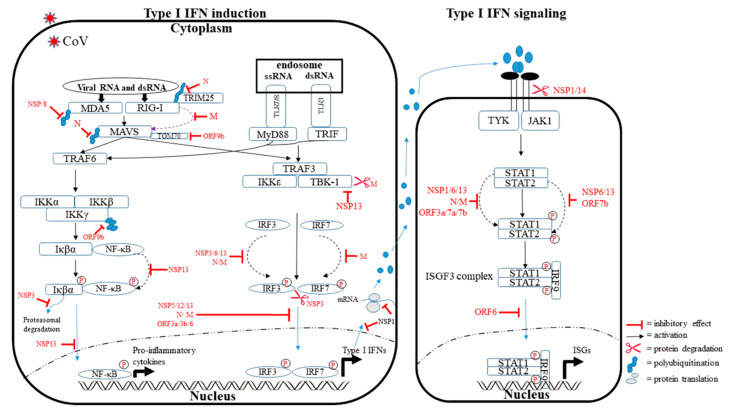
Type I IFN production and signaling pathway suppressed by SARS-CoV-2: Upon viral infection, the recognition by host innate immune sensors such as RLRs and TLRs triggers an antiviral signaling cascade, resulting in the production of IFNs and the activation of IFN signaling pathways, eventually ensuing in the production of ISGs. In contrast, SARS-CoV-2 has developed strategies to counteract the host defense system by inhibiting protein–protein interactions (dotted purple lines), the phosphorylation of proteins (dotted black lines), degrading of host cellular proteins (scissors), and cellular localization of proteins (blue arrowed lines). DMV: double-membrane vesicles.

**Figure 3 viruses-14-00530-f003:**
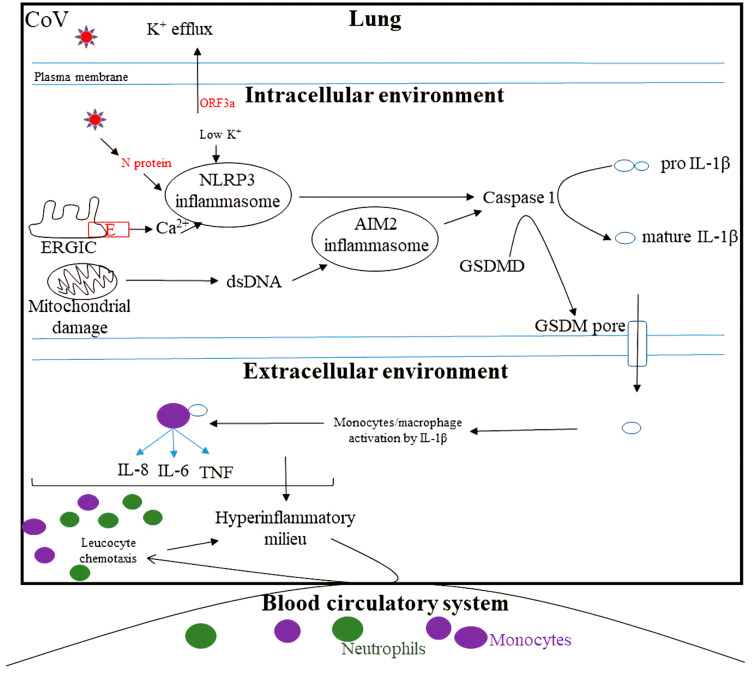
Activation of inflammasome by CoV, leading to hyperinflammation: The viroporins, ORF3a, and E can trigger K^+^ efflux or Ca^2+^ influx to induce NLRP3 activation. Moreover, the N protein can bind directly to NLRP3, inducing its activation. Additionally, mitochondrial damage can result in the production of dsDNA, which activates the AIM2 inflammasome. NLRP3 and AIM2 inflammasomes activate caspase 1, which cleaves GSDMD into amino-terminal and carboxy-terminal fragments. The GSDMD amino-terminal fragment targets and inserts itself into cellular membrane lipids, forming pores and rendering the membranes permeable. Caspase 1 cleaves pro-IL-1β into mature IL-1β, which is released through the GSDMD pore. IL-1β can stimulate monocytes to secrete additional proinflammatory cytokines such as IL-6, IL-8, and TNFα, which further upregulate inflammation by recruiting lymphocytes such as neutrophils.

**Table 1 viruses-14-00530-t001:** Mechanism by which CoV proteins evade recognition by innate immune sensors and inhibit INF production and signaling pathways.

SARS-CoV-2 Protein	Inhibitory Effect on Immune System	Mechanism of Action	Reference
NSP1	mRNA export	Prevention of adequate NXF1 and host mRNA interaction	[55]
	Protein translation	Interaction with the 40S ribosomal subunit	
	Inhibits Type I IFN signaling pathway	Inhibition of the phosphorylation of STAT1 and STAT2	
NSP3	Recognition by PRRs	DMV formation, helping to escape recognition of vRNA by molecular sensors	[46,47,48,63]
	Protein degradation	Cleavage of IRF3 through papain-like protease activity	
	Type I IFN synthesis	Prevention of IRF3 phosphorylation, dimerization, and translocation	
NSP4	Recognition by PRRs	DMV formation	[46,47,48]
NSP6	Recognition by PRRs	DMV formation	[46,47,48,68]
	Type I IFN synthesis	Inhibition of IRF3 phosphorylation	
	Type I IFN signaling pathway	Inhibition of STAT1 and STAT2 phosphorylation	
NSP8	Type I IFN synthesis	Impairment of K63-linked polyubiquitination of MDA5	[61,71]
	Protein trafficking	Binding to 7SLRNA	
NSP9	Protein trafficking	Binding to 7SLRNA	
NSP12	Type I IFN signaling pathway	Inhibition of the nuclear translocation of IRF3	[72]
NSP13	Recognition by PRRs	Hydrolysis of the first phosphate from the nascent RNA	[49,68,73,74]
	Type I IFN synthesis	Down-regulation of TBK1 recruitment to MAVS	
	Type I IFN signaling pathway	Inhibition of STAT1 and STAT2 phosphorylation	
NSP14	Recognition by PRRs	Methylation of capped viral RNA together with NSP10	[46,50]
	Protein degradation	Lysosomal degradation of IFNAR1	
NSP16	Recognition by PRRs	Methylation of the ribose 2′-O of the first nucleotide of viral Cap-O-RNA together with NSP10	[51,52,61]
	Decreased ISG activity	Disruption of host mRNA splicing	
N	Type I IFN synthesis	Inhibition of TRIM25-mediated RIG-I ubiquitination and activationand inhibition of TBK1-IRF3 interaction, phosphorylation, and nuclear translocation of IRF3	[76,77,78,80]
	Type I IFN signaling pathway	Inhibition of the phosphorylation and nuclear translocation of STAT1 and STAT2	
M	Protein degradation	Proteasomal degradation of TBK1 through K48-linked ubiquitination	[62,68,83]
	Type I IFN synthesis	Inhibition of RIG-I—MAVS, MAVS—TBK1, TRAF3—TBK1 interactions. Downregulation of phosphorylation and nuclear translocation of IRF3	
ORF3a	Type I IFN synthesis	Inhibition of IRF3 nuclear translocation	[68,74]
	Type I IFN signaling pathway	Inhibition of STAT1 phosphorylation	
ORF3b	Type I IFN synthesis	Inhibition of IRF3 nuclear translocation by sequestering IRF3 outside the nucleus	[86]
ORF6	Type I IFN synthesis	Inhibition of IRF3 nuclear translocation	[68,88,89,90]
	Type I IFN signaling pathway	Inhibition of STAT1 nuclear translocation	
ORF7a	Type I IFN signaling pathway	Inhibition of STAT1 phosphorylation	[68]
ORF7b	Type I IFN signaling pathway	Inhibition of STAT1 and STAT2 phosphorylation	[68]
ORF9b	Type I IFN synthesis	Interruption of K63-linked ubiquitination of IKKγInteraction with TOM70, presumably disrupting MAVS and TBK1/IRF3 interaction	[25,94]

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
