# Peer review of "Escape and Over-Activation of Innate Immune Responses by SARS-CoV-2: Two Faces of a Coin"

_viruses, 2022, doi:10.3390/v14030530_

Round 1

Reviewer 1 Report

Review by Mattoo et al., describes the diverse strategies developed by coronaviruses including SARS-CoV-2 to evade from innate immune recognition and signaling, and explains how dysregulation of viral sensing contributes to pathogenesis. The review is well written, interesting for the field, and provide a comprehensive overview of the various mechanisms developed by coronaviruses to inactivate sensing/signaling pathways.

Here are some concerns:

A effort would be appreciated to cite original research articles rather than previous review on the same topic. For example, ref [30] is citated 4 times and could be replaced by original research articles.

Some citations are misleading:

It is written: “In contrast, activation of inflammasomes by SARS-CoV-2 initiates an inflammatory form of cell death, triggering the release of proinflammatory cytokines IL-1β and IL-18 [5].” This sentence is misleading because reference [5] link to a review on the inflammasome topic but unrelated to SARS-CoV-2. Please remove “SARS-CoV-2” from this sentence or cite a SARS-CoV-2 related paper.

“Uncontrolled inflammasome activation has been associated with COVID-19 disease severity [6-8].” Reference [7] is not related to SARS-CoV-2, please correct this point.

Some citations are missing:

Only one citation refers to SARS-CoV-2 detection by MDA-5. The role of MDA-5 as sensor of SARS-CoV-2 has been identified by several teams please cite accordingly (ex doi.org/10.1038/s41598-021-92940-3, and doi.org/10.1128/JVI.02415-20 in addition to DOI: 10.1016/j.celrep.2020.108628).

Please complete the (Ref) mark with appropriate citations that are missing to support the statement: “PRRs, such as DExD/H-box RNA helicase family members DHX33 and DDX19A, have been reported to interact with and activate NLRP3 after sensing reoviral genomic RNA (Ref). Moreover, the roles of ZBP1, RIG-I, and TLR3 have also been shown to be involved in driving the activation of NLRP3 inflammasome (Ref).”

“Overexpression of SARS-CoV-2 ORF3a caused K+, NLRP3 activation, and IL-1β release in the lung epithelial cell line A549 (Ref). Among structural proteins, SARS-CoV-2 E protein has been observed to function as a K+ -permeable cation channel, lysing cells and causing ARDS-like disease in mice (Ref).”

Some recent point could be discussed to strengthen the manuscript:

SARS-CoV-2 also developed strategies to escape for inflammasome activation. For instance, recent work highlighted that SARS-CoV-2 inhibit proptosis via its nucleocapsid protein N (DOI 10.15252/embj.2021108249). Discussion of such mechanisms developed by SARS-CoV-2 to inhibit or overactivate inflammasome would strengthen the manuscript.

Author Response

We appreciate reviewer 1’s constructive and helpful comments and suggestions. We believe that our manuscript is now much improved by addressing his/her opinions. Below, we provide line-by-line authors’ responses.

Reviewer #1

Review by Mattoo et al., describes the diverse strategies developed by coronaviruses including SARS-CoV-2 to evade from innate immune recognition and signaling, and explains how dysregulation of viral sensing contributes to pathogenesis. The review is well written, interesting for the field, and provide a comprehensive overview of the various mechanisms developed by coronaviruses to inactivate sensing/signaling pathways.

Here are some concerns:

A effort would be appreciated to cite original research articles rather than previous review on the same topic. For example, ref [30] is citated 4 times and could be replaced by original research articles.

We are thankful to reviewer 1 for important suggestion. In the revised manuscript we have replaced reference [30] by original research articles. 

Some citations are misleading:

It is written: “In contrast, activation of inflammasomes by SARS-CoV-2 initiates an inflammatory form of cell death, triggering the release of proinflammatory cytokines IL-1β and IL-18 [5].” This sentence is misleading because reference [5] link to a review on the inflammasome topic but unrelated to SARS-CoV-2. Please remove “SARS-CoV-2” from this sentence or cite a SARS-CoV-2 related paper.

We are thankful to reviewer 1 for careful reading. We added the relevant references defining inflammasome activation by SARS-CoV-2 in the revised manuscript, as follows:

In contrast, activation of inflammasomes by CoV initiates an inflammatory form of cell death, triggering the release of proinflammatory cytokines IL-1β and IL-18 [5-9]

“Uncontrolled inflammasome activation has been associated with COVID-19 disease severity [6-8].” Reference [7] is not related to SARS-CoV-2, please correct this point.

We are thankful to reviewer 1 for careful reading. We have removed reference [7] in the revised manuscript, as follows:

Uncontrolled inflammasome activation has been associated with disease severity [10].

Some citations are missing:

Only one citation refers to SARS-CoV-2 detection by MDA-5. The role of MDA-5 as sensor of SARS-CoV-2 has been identified by several teams please cite accordingly (ex doi.org/10.1038/s41598-021-92940-3, and doi.org/10.1128/JVI.02415-20 in addition to DOI: 10.1016/j.celrep.2020.108628).

We are thankful to reviewer 1 for providing help. We have cited all the above references in the revised manuscript. References [21-24]

Please complete the (Ref) mark with appropriate citations that are missing to support the statement: “PRRs, such as DExD/H-box RNA helicase family members DHX33 and DDX19A, have been reported to interact with and activate NLRP3 after sensing reoviral genomic RNA (Ref). Moreover, the roles of ZBP1, RIG-I, and TLR3 have also been shown to be involved in driving the activation of NLRP3 inflammasome (Ref).”

Thank you very much for suggestion. We have added the references in the revised manuscript as follows:

PRRs, such as DExD/H-box RNA helicase family members DHX33 and DDX19A, have been reported to interact with and activate NLRP3 after sensing reoviral genomic RNA [39]. Moreover, the roles of ZBP1, RIG-I, and TLR3 have also been shown to be involved in driving the activation of NLRP3 inflammasome [39].

“Overexpression of SARS-CoV-2 ORF3a caused K+, NLRP3 activation, and IL-1β release in the lung epithelial cell line A549 (Ref). Among structural proteins, SARS-CoV-2 E protein has been observed to function as a K+ -permeable cation channel, lysing cells and causing ARDS-like disease in mice (Ref).”

Thank you very much for suggestion. We have added the references in the revised manuscript as follows:

Line 334-337

Overexpression of SARS-CoV-2 ORF3a caused K+ efflux, NLRP3 activation, and IL-1 β release in the lung epithelial cell line A549 [96]. Among structural proteins, SARS-CoV-2 E protein has been observed to function as a K+-permeable cation channel, lysing cells and causing ARDS-like disease in mice [97].

Some recent point could be discussed to strengthen the manuscript:

SARS-CoV-2 also developed strategies to escape for inflammasome activation. For instance, recent work highlighted that SARS-CoV-2 inhibit proptosis via its nucleocapsid protein N (DOI 10.15252/embj.2021108249). Discussion of such mechanisms developed by SARS-CoV-2 to inhibit or overactivate inflammasome would strengthen the manuscript.

We are thankful to reviewer 1 for giving an important suggestion. We added the relevant article showing the role of nucleocapsid protein N in inhibiting pyroptosis. We have added the relevant information in the revised manuscript as follows:

Line 358-364

In contrast, Nucleocapsid binding to GSDMD was shown to suppress pyroptosis, preventing immunological signaling components from being secreted to the outside [104]. For example, GSDMD inhibition allowed host cells to accumulate large amounts of mature IL-1 in the cytoplasm, resulting in the release of a large amount of inflammatory cytokines in the lytic phase [104]. This strategy might be helping the virus to evade immune assault at early stages of infection and eventually resulting in a large quantities of cytokine release from dying monocytes at later stages which would promote hyperinflammation and cytokine storm in patients [104,105].

Reviewer 2 Report

In this review by Mattoo et al., the authors discuss how Coronaviruses such as SARS-CoV1, CoV2, and MERS-CoV can escape the innate immune signaling of the host after infection. This causes the failure of the host to control the infection in the initial stage (innate), resulting in an exacerbated pathogenesis. The authors also discuss the activation of inflammasomes because of the infection, which though important, when uncontrolled, can cause a huge surge in the production of inflammatory cytokines that can result in severe cytokine storms that prove to be fatal.

Major comments:

  1. Since the introduction mostly covers the comparison of the genomic structure of the three coronaviruses, a figure showing the genomic structure of the three different viruses will help make things easy to understand.
  2. Figure 1 is very congested and difficult to follow; please simplify the cartoon.
  3. The authors do not mention the cGAS-STING pathway under host immune sensors; studies have shown it to be activated by SARS-CoV2 (e.g., PMID: 34732709)
  4. Unlike RLR and TLR3, the SARS-CoV2 context of inflammasomes is missing under the ‘host innate immune sensors’ section.
  5. Ln 199: The authors do not cover NSP4’s role in escaping innate immune response; however, they mention it in the title of the section.
  6. Although the authors claim to cover MERS-CoV as one of the viruses in the abstract, there’s very little information about it overall in the review- almost negligible under the inflammasome section.
  7. Please add a section on how these pathways can be targeted therapeutically for treatment against SARS-CoV-2 infection.

Minor comments:

  1. Ln 83: IFN with apostrophe s (‘s) does not make sense in the sentence.
  2. The conclusion can be a little more elaborate to cover more aspects.

Author Response

We appreciate the reviewer 2’s constructive and helpful comments and suggestions. We believe that our manuscript is now much improved by addressing his/her opinions. Below, we provide line-by-line authors’ responses.

Reviewer #2

In this review by Mattoo et al., the authors discuss how Coronaviruses such as SARS-CoV1, CoV2, and MERS-CoV can escape the innate immune signaling of the host after infection. This causes the failure of the host to control the infection in the initial stage (innate), resulting in an exacerbated pathogenesis. The authors also discuss the activation of inflammasomes because of the infection, which though important, when uncontrolled, can cause a huge surge in the production of inflammatory cytokines that can result in severe cytokine storms that prove to be fatal.

Major comments:

  1. Since the introduction mostly covers the comparison of the genomic structure of the three coronaviruses, a figure showing the genomic structure of the three different viruses will help make things easy to understand.

We are thankful to reviewer 2 for suggesting a figure showing the genomic structure of the three coronaviruses. Please refer to figure 1 in the revised manuscript.

  1. Figure 1 is very congested and difficult to follow; please simplify the cartoon.

We have simplified the Figure 1, now Figure 2 in the revised manuscript.

  1. The authors do not mention the cGAS-STING pathway under host immune sensors; studies have shown it to be activated by SARS-CoV2 (e.g., PMID: 34732709)

We understand reviewer’s concern and are thankful for the suggestion. However, cGAS-STING will make the manuscript very lengthy and is out of our scope.

  1. Unlike RLR and TLR3, the SARS-CoV2 context of inflammasomes is missing under the ‘host innate immune sensors’ section.

As the reviewer suggested, we have rearranged the manuscript and moved the related part, from the section ‘Inflammasome activation by CoV’, describing inflammasome sensors to the section ‘host innate immune sensors’ in the updated manuscript. Line 117-127.

  1. Ln 199: The authors do not cover NSP4’s role in escaping innate immune response; however, they mention it in the title of the section.

We are thankful to reviewer 2 for pointing it out. We have deleted NSP4 from the title. Instead, we have discussed it in the section ‘sheltering and modifying viral RNA to evade PRR recognition’.

  1. Although the authors claim to cover MERS-CoV as one of the viruses in the abstract, there’s very little information about it overall in the review- almost negligible under the inflammasome section.

 We agree that we haven’t described the functions of MERS-CoV-encoded genes in the inflammasome section. As sufficient evidence has been accumulated on the role of MERS-CoV proteins in the dysregulation of IFN signaling, we rather decided to focus on the role of MERS-CoV proteins in inhibiting IFN expression and signaling.

  1. Please add a section on how these pathways can be targeted therapeutically for treatment against SARS-CoV-2 infection.

We are thankful to reviewer 2 for suggesting to add a section summarizing the therapeutics against COVID-19. We have added a section as follows:

Line 406-440.

Section ‘Therapeutics targeting innate immune system’. Line 373-406.

           During the COVID-19 pandemic, various treatment strategies have been investigated, including antiviral therapies and inflammatory cytokine antagonists.

           Antivirals including remdesivir, molnupiravir, paxlovid, and anti-spike monoclonal antibodies including casirivimab–imdevimab, bebtelovimab, and sotrovimab have been approved by FDA or have been given EUA [107]. In addition, antiviral therapies including type I and type III IFN treatments modulate immune cell activation and inflammation have also been explored. For example, a preclinical study showed that exogenous type I IFN therapy prior to infection reduced the viral load [108]. However, the effect was limited once the infection was established. Similarly, a study showed that administration of IFN-α2b during the early phase of COVID-19 (within 5 days post-hospitalization) was associated with a decreased mortality [109]. However, later administration of IFN-α2b was associated with an increase in mortality. In addition, a RIG-I agonist, stem-loop RNA 14 prevented viral infection in the lower respiratory tract and the development of severe disease in a mouse model of SARS-CoV-2 infection [110]. Similarly, like type I IFNs, it is well established that type III IFNs are the first-line defense against viral infections [111]. SARS-CoV-2 infection was reduced when human Calu-3 cells, intestinal epithelial cells, and colon organoids were pretreated with IFN- λ while the absence of the type III IFN receptor increased susceptibility to infection [112-114]. Similarly, in mice, administration of pegylated (PEG)- IFN-λ1a diminished SARS-CoV-2 replication [114]. Interestingly, in observational studies, increased IFN-λ1 and IFN-λ2 levels in serum were associated with a better prognosis [115,116]. Based on these data, clinical trials have begun to evaluate the effect of PEG-IFN-λ1a therapy in COVID19 patients, which have shown contradictory results. In a phase 2 trial, patients with the mild disease showed no significant differences in time to symptom resolution or other clinical metrics between the PEG-IFN-λ1a and placebo groups [117]. However, another phase 2 trial study reported that PEG-IFN-λ1a decreased the levels of in mid-turbinate viral load by day 7 [118].

In addition, inflammatory cytokine agonists such as anti-IL-1 and anti-IL-6 therapies have been evaluated. For example, with antagonists to IL-1 or its receptor, results are more or less contradictory: 1) an observational study found that in COVID-19 hospitalized patients, two doses of canakinumab, an anti-IL-1β antibody, can lead to clinical improvement while a single dose was reported to be ineffective in a phase 3 study [119,120]. 2). Anakinra, an IL-1 receptor antagonist, was shown ineffective in earlier studies [121]. While another studies observed a reduction in inflammatory markers and mortality in patients with COVID-19 [122,123]. To resolve apparent controversy in those studies, a more in-depth analysis of cytokine signaling inhibitors is warranted.

Reviewer 3 Report

This review properly summarizes the innate immune system involved in SARS-CoV-2. I approve this report for publication.

Author Response

We appreciate the reviewer 3’s enthusiasm about our manuscript. Thank you very much indeed!